# Design Guidelines for Thin Diaphragm-Based Microsystems through Comprehensive Numerical and Analytical Studies

**DOI:** 10.3390/mi14091725

**Published:** 2023-09-01

**Authors:** Vinod Belwanshi, Kedarnath Rane, Vibhor Kumar, Bidhan Pramanick

**Affiliations:** 1CSIR-National Metallurgical Laboratory, Jamshedpur 831007, India; 2School of Physics and Astronomy, University of Glasgow, Glasgow G12 8QQ, UK; 3National Manufacturing Institute Scotland Renfrew, Renfrew PA4 9PA, UK; kedarnath.rane@strath.ac.uk; 4School of Engineering, Rutgers, The State University of New Jersey, Piscataway, NJ 08854, USA; 5School of Electrical Sciences, Center of Excellence in Particulates Colloids and Interfaces, Indian Institute of Technology Goa, Ponda 403401, India

**Keywords:** design and analysis, finite element analysis, microsystems, MEMS, thin diaphragm

## Abstract

This paper presents comprehensive guidelines for the design and analysis of a thin diaphragm that is used in a variety of microsystems, including microphones and pressure sensors. It highlights the empirical relations that can be utilized for the design of thin diaphragm-based microsystems (TDMS). Design guidelines developed through a Finite Element Analysis (FEA) limit the iterative efforts to fabricate TDMS. These design guidelines are validated analytically, with the assumption that the material properties are isotropic, and the deviation from anisotropic material is calculated. In the FEA simulations, a large deflection theory is taken into account to incorporate nonlinearity, such that a critical dimensional ratio of a/h or 2r/h can be decided to have the linear response of a thin diaphragm. The observed differences of 12% in the deflection and 13% in the induced stresses from the analytical calculations are attributed to the anisotropic material consideration in the FEA model. It suggests that, up to a critical ratio (a/h or 2r/h), the thin diaphragm shows a linear relationship with a high sensitivity. The study also presents a few empirical relations to finalize the geometrical parameters of the thin diaphragm in terms of its edge length or radius and thickness. Utilizing the critical ratio calculated in the static FEA analysis, the basic conventional geometries are considered for harmonic analyses to understand the frequency response of the thin diaphragms, which is a primary sensing element for microphone applications and many more. This work provides a solution to microelectromechanical system (MEMS) developers for reducing cost and time while conceptualizing TDMS designs.

## 1. Introduction

The continuously increased demand for microsystems, such as microphones and pressure sensors, in various applications such as the consumer [1,2], medical [3,4], automotive [5] industries, and many more, has forced incredible development in the field of TDMS using MEMS technology. MEMS aim for all purposes, where a tiny size, high-level sound quality, reliability, and affordability are the key requirements. MEMS technology reduces the production cost per device, makes batch fabrication possible, provides a compact size, and offers an ease of implementation for microsystems. It also offers low power consumption, a good sensitivity, and is available in very small packages that are fully compatible with surface mount assembly processes [6,7,8,9,10]. There are standard micro elements, such as a thin diaphragm, cantilever, and suspended diagrams, etc., that are utilized in microsystems or MEMS [11]. However, TDMS are found to be more robust and reliable, along with having an ease of fabrication and higher sensitivity. Microphones and pressure sensors typically use a thin diaphragm, and can be classified as a TDMS. The tale of MEMS microphones began in 1988, when Knowles Electronics fabricated silicon-based microphones suitable for the hearing aid industry [12]. The majority of their microphones are currently utilized in mobile phones, digital cameras, computers, MP3 players, tablets, laptops, smart televisions, automotive voice recognition, gaming and remote controllers, and so on [6,13]. The diaphragm vibrates when it is subjected to a pressure variation that occurs due to an incoming soundwave. The vibration of the thin diaphragm modulates the capacitor, and these modulations are amplified further using suitable signal conditioning [14,15]. The key advantage of this microphone is that it does not lose any sensitivity over time or under elevated temperatures due to no electret material being used [16,17]. A comprehensive review of MEMS microphones was recently published and covered aspects such as materials, fabrication methods, and performances [7]. This work gave a clear direction for the applicability of microphones. MEMS pressure sensors are one of the most reported and developed microsystems that also utilize a thin diaphragm. Pressure sensors typically the measure deflection/deformation of a thin square or circular or rectangular diaphragm under applied external pressure. This mechanical deflection can be converted into an electrical signal by the means of various transduction techniques [11,18,19]. MEMS pressure sensor commercialization was started in the 1970s and 1980s by various companies (e.g., Foxboro ICT, Transensory Devices, IC Sensors, IC Transducers, Kulite Inc., and Novasensor), which used silicon diaphragms and are considered to be one of the earliest commercial successes of microsystems devices [20,21]. Thereafter, the design and development of such microsystems for various applications have gained a tremendous pace and resulted in the development of various TDMS. However, before implementing the MEMS fabrication of such microsystems, an iterative process of design optimization is needed [22]. A finite element analysis (FEA), through various FEA tools such as ANSYS, COMSOL Multiphysics, and Coventorware, is useful for limiting this iterative design campaign and saves on the overall product cost. The design of diaphragm-based microsystems was analyzed through numerical, analytical, and experimental investigations. These studies include the vibration of square plates [23], spring-supported capacitive designs [24], simply supported rectangular plates [25], moderately thick rectangular plates [26], biomimetic directional designs [27], a single deeply corrugated diaphragm technique [28], thin plates [29], simply supported square/rectangular plates [19,30], circular plates in axisymmetric modes [31], the unique directional properties of dual diaphragms [32], rectangular plates using Bessel functions [33], and the free vibration of square plates [34]. The current interest in microsystems is to develop an application-oriented design with a better performance using thin-diaphragm configurations. There are several studies on the design and development of microsystems for different shapes of diaphragm, including circular and rectangular geometries. The aspect ratios of these geometries range from 200 to 2000. For an example, microphones’ sensitivity to this aspect ratio falls between a few 10s of µV/Pa and 10s of mV/Pa [35,36,37,38,39,40,41]. However, this reported sensitivity is influenced by various parameters, including the supplied voltage, transaction techniques, and other electrical inputs. These studies use thin diaphragms and require estimating their mechanical sensitivity as a preliminary design consideration for designing diaphragm-based microsystems. This is because the mechanical responses of a diaphragm act as a primary sensing element.

This paper provides the design, simulation, and analytical validation of a TDMS for a generalized approach to TDMS design. Additionally, this paper facilitates achieving an optimized design parameter through a defined critical ratio for a square and circular diaphragm as a/h or 2r/h,respectively. Furthermore, the role of the aspect ratio can be introduced in this primary sensing element design through establishing a generalized empirical relationship for providing a TDMS with optimized design parameters. Furthermore, based on selected responses, such as deflection and induced stresses, a feasible design window for the TDMS is proposed for a higher sensitivity. The optimized design parameters are also verified through a static analysis in dynamic load cases through a harmonic analysis to understand the frequency response. The paper is divided as follows: Section 2 deals with the operating principle of the thin diaphragm used in the MEMS, Section 3 explains the materials and methodology used to complete the current research work, Section 4 elaborates the results obtained and the discussion, and Section 5 outlines the conclusions.

## 2. Principle of Operation

The operation of a TDMS depends on the change in deflection or induced stress due to an externally applied load or pressure. The TDMS consists of a thin diaphragm as a primary sensing element that deflects (or induces stress) under the pressure change caused by the movement of physical media such as air. The deflection or induced stress can be converted using different transduction techniques, such as capacitive or piezoresistive techniques. In order to design a TDMS, the selection of the optimal geometrical parameters is very crucial in deciding the overall sensitivity of such a microsystem. In the case of capacitive or piezoresistive transduction techniques, the deflection or induced stress, respectively, should be as large as possible for the given geometry of the thin diaphragm. This decides the mechanical sensitivities in terms of the deflection per applied pressure or induced stress per applied pressure for the thin diaphragm. Further, based on the transduction techniques, the overall sensitivity is calculated in terms of the output voltage or current. This electrical output can be amplified and further processed using a suitable signal-conditioning circuit. A general scheme for the operation of a TDMS is explained using Figure 1.

Mostly, square or circular thin diaphragms are used for microsystems and their geometrical parameters are crucial to obtaining a higher sensitivity. If a square thin diaphragm with an edge length of 2a and thickness of h has all its edges fixed and a pressure of p is applied to it, the simplified deflection and induced stress can be defined as below [19]:

Normalized deflection with the thickness of a thin diaphragm:(1)$$\frac{d}{h}={\frac{\left(1-\vartheta^{2}\right)}{4E}\left(\frac{a}{h}\right)}^{4}p\;\mathrm{or}\;\frac{d}{h}=c_{ds}{\left(\frac{a}{h}\right)}^{4}$$

Induced stress can be given by:(2)$$\sigma_{sl}={\left(\frac{a}{h}\right)}^{2}p\;\mathrm{and}\;\sigma_{st}={\vartheta \left(\frac{a}{h}\right)}^{2}p$$

Similarly, a circular plate deflection and induced stress can be defined as below [42]:

Normalized deflection with the thickness of a thin diaphragm:(3)$$\frac{d}{h}={\frac{3\left(1-\vartheta^{2}\right)}{16E}\left(\frac{2r}{h}\right)}^{4}p\;\mathrm{or}\;\frac{d}{h}=c_{dc}{\left(\frac{2r}{h}\right)}^{4}$$

The maximum induced stress can be given by:(4)$$\sigma_{cr}={\frac{3}{4}\left(\frac{2r}{h}\right)}^{2}p\;\mathrm{and}\;\sigma_{ct}={\frac{3}{4}\vartheta \left(\frac{2r}{h}\right)}^{2}p$$
where E is the Young’s modulus, ϑ is the Poisson’s ratio, d is the deflection, h is the thickness, a is the edge length, and r is the radius of the thin diaphragm.

## 3. Materials and Methodology

A simplified view of square- and circular-shaped thin diaphragms, as presented in Figure 1a,b, with a 2a edge length and r radius, respectively, and h thickness, was used in this study. The FEA model was validated against the simplified equations before performing the parametric analysis using isotropic silicon material properties. However, the anisotropic behavior of silicon is mandatory to be considered for the estimation of the actual response of the thin diaphragm; therefore, anisotropy material properties were included in the model and the deviation from the isotropic behavior of silicon was observed. To realize a thin diaphragm using MEMS technology, a general consideration is that a silicon-based thin diaphragm has a slanted cut if wet chemistry (KOH or TMAH etch) is used, as shown in Figure 1c, or a straight cut if dry chemistry is used, as shown in Figure 1d. The silicon material properties listed in Table 1 were used for the study. The methodology was to simulate the isotropic silicon material and its analytical validation, thereafter anisotropic silicon properties were utilized in the model. The responses were analyzed and are presented in the next section. Additionally, the actual fabricated geometry was also analyzed to understand the deviation from the ideal diaphragms.

## 4. Results and Discussions

In the current section, the results of the FEA simulation study and its analytical validations are presented for the square and circular diaphragms. As can be seen in Figure 2, the maximum deflection for the thin diaphragm was observed in its center, irrespective of its shape. It can also be observed that the maximum induced stress was in the edge of the diaphragms. Contour plots are presented in Figure 2 to provide a comparative qualitative visualization of the deflection and induced stress on the thin diaphragm. Extensive quantitative analyses are also presented in further detail.

### 4.1. Model Validation

The FEA model was validated against the simplified analytical model, as discussed in the earlier section. A square thin diaphragm with dimensions of 500 µm × 500 µm × 10 µm was chosen to validate the FEA model with the simplified analytical equations. As the analytical simplified equations utilize isotropic material properties, the FEA was carried out with the isotropic silicon material properties and the response in terms of the deflection and induced stresses can be seen from Figure 3. The deflection and induced stresses for the analytical model and FEA that utilizes isotropic material showed a very good agreement with each other. The percentage of changes in the deflection and induced stress showed 3.48% and 2.27%, respectively, for the FEA model and analytical calculations. When the anisotropic material properties of silicon were included in the model, then the deviations from the analytical calculation were observed to be 12.81% and 13.01% for the deflection and induced stresses, respectively. As the model was validated against the isotropic material properties, it can be concluded that the model, which utilized anisotropic material properties, would also give the appropriate response. Parametric analyses were further performed to optimize the geometry and are presented in the next subsection. Figure 3a,b shows the pressure response of the thin diaphragm in terms of the deflection and induced stresses, respectively. The deviation from the analytical calculations can be seen from Figure 3 and is qualified in Table 2. It also explains the deviation from the isotropic model to the anisotropic model with the wet- (KOH/TMAH) and dry-etched geometry of the thin diaphragm.

### 4.2. Parametric FEA Analysis of Thin Diaphragms/Plates

In this parametric analysis, the anisotropic material properties of silicon were utilized. These simulations were carried out under the consideration of a large deflection that accounted for structural nonlinearity during the simulations. The FEA results suggest that there was a critical limitation of the lateral dimensions and thickness of the thin diaphragm. As soon as these dimensions were exceeded, the response of the thin diaphragm (in terms of the deflection and induced stress) did not follow the simplified analytical calculation and accounted for the nonlinear behavior with the applied load. In order to validate this, a parametric analysis was performed with a varying a/h ratio of the diaphragm. As can be seen from Figure 4, there was an increase in the deflection at a higher rate up to the critical a/h ratio, thereafter, both the deflection of the diaphragm decreased at a slower rate of increase in the deflection as a function of the a/h or 2r/h ratio. The parametric results are presented for the square and circular diaphragms in Figure 4a and Figure 4b, respectively.

Since the curve is plotted in a log-log graph, it is difficult to read a small difference between the square and circular thin diaphragms’ deflection. Hence, the normalized deflections (d/h) were analyzed for a/h and 2r/h. As d/h is a function of the ${\left(a/h\right)}^{4}$ or ${\left(2r/h\right)}^{4}$, the proportionality constants were calculated as 1.9 × 10^−12^ and 1.5 × 10^−12^ for the square and circular thin diaphragms, respectively, under a 20 Pa applied pressure. It can be observed from Figure 4 that, up to a certain ratio of a/h or 2r/h., the d/h followed the ${\left(a/h\right)}^{4}$ or ${\left(2r/h\right)}^{4}$ with a higher sensitivity, and afterwards, it started deviating and showed a lower sensitivity, even for a thinner diaphragm. This ratio is named as the critical ratio and it was observed that, if the deflection of a thin diaphragm is beyond 1/5th of its thickness, it starts to decrease in sensitivity (Figure 5). The calculated ratios ${\left(a/h\right)}^{4}$ and ${\left(2r/h\right)}^{4}$ were ~569.6 and 604.3 for the square and circular thin diaphragms, respectively, at d/h = 1/5.

Similarly, the induced stress, as shown in Figure 6, is a function of ${\left(\frac{a}{h}\right)}^{2}$ or ${\left(\frac{2r}{h}\right)}^{2}$ and the proportionality constants were calculated as 5.45 × 10^−6^ and 3.21 × 10^−6^ for the square and circular thin diaphragms, respectively, under the 20 Pa applied pressure. At the critical $\frac{a}{h}$ (569.6), the induced stress was 1.77 MPa. Likewise, for the circular plate, a critical $\frac{2r}{h}$ (605) resulted in a 1.17 MPa induced stress.

The frequencies of the thin diaphragms were simulated with a varying thickness or a/h ratio, as shown in Figure 7a. As can be seen, the frequency of the thin diaphragm was strongly dependent on the a/h ratio, as well as the lateral dimension of the diaphragm. To establish an empirical relation between the natural frequency and geometrical ratio (a/h or 2r/h), as shown in Figure 7b, the thickness was used as a multiplier for the frequency and the proportionality coefficient was calculated. Distinct coefficients were calculated as 1.12 × 10^10^ and 1.45 × 10^10^ for the square and circular plates, respectively (Figure 7b).

The empirical relationship established and its coefficients for deflection, induced stress, and factor of natural frequency are presented in Table 3. The table provides a comprehensive guideline for the critical parameters of a TDMS.

It was observed that a deflection of up to 20% of the thin plate thickness followed the simplified analytical relation, which did not account for the nonlinearity due to the material and geometry. Based on this assumption, the critical ratios for the square and circular thin diaphragms/plates were calculated as 570 and 605, respectively. Thereafter, the edge of the square diaphragm and radius of the circular plate were calculated (and are presented in Table 4) as 1425 µm and 756.25 µm, respectively, for a 2.5 µm thickness of the thin plate. The deflection and induced stress are plotted under the applied load, as shown in Figure 8.

### 4.3. Harmonic Analysis

A harmonic analysis of the thin diaphragm/plate was performed and analyzed. At the resonant frequency (17.2 kHz) of the thin diaphragm, it gave infinite deflection. To bring it to a practical deflection value, a small damping ratio (0.07) was included during the numerical simulation. It can be seen that a peak at the resonant frequency (17.2 kHz) of the plate was observed for the maximum deflection and stresses.

The deflection and induced stresses, as a function of the applied frequency and applied load, are presented in Figure 9a,b. The thin diaphragm had a resonant frequency of 17.2 kHz. Also observed was that the responses were dependent on the applied pressure as well. It can be observed that the frequency response was flat up to 4 kHz with the pressure ranging from 5 Pa to 20 Pa, showing linearity between the applied pressure and deflection until a high mid-range of audio signal (4 kHz). Beyond 4 kHz (presence and brilliance range), a sudden jump in the deflections was observed due to attaining the resonant frequency of 17.2 kHz.

## 5. Conclusions

This paper demonstrated the design, simulation, and analytical validation of MEMS, particularly TDMS, towards a generalized approach using the FEA-based technique. It facilitated achieving an optimized design parameter through a defined critical ratio for square and circular diaphragms as a/h or 2r/h,respectively. The calculated critical ratios, *a/h* = 570 for the square and *2r/h* = 605 for the circular diaphragms, were validated through simplified analytical models. These ratios are helpful for calculating the geometrical parameters of thin diaphragms that have linear responses. A few empirical relations were developed for approximating the deflection, induced stresses, and natural frequencies in the thin diaphragm, which act as a convenient solution for defining the limits of the geometric parameters of a TDMS. It was observed that the responses in terms of the deflections and induced stresses depended on the geometrical parameters of the thin diaphragm. The optimized design parameters, through a static analysis, were also tested in dynamic load cases through a harmonic analysis to understand the frequency response.

## Figures and Tables

**Figure 1 micromachines-14-01725-f001:**
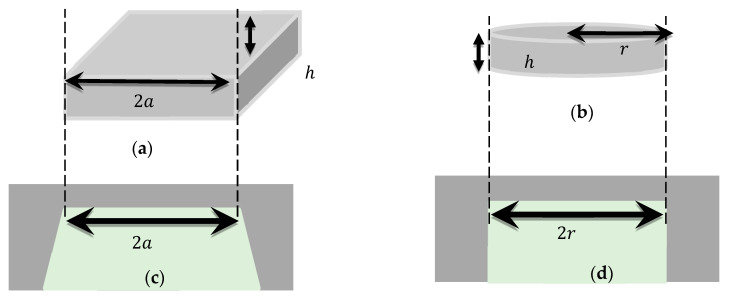
The schematics of simplified thin diaphragms, (**a**) square and (**b**) circular, used in the analysis. (**c**) Wet and (**d**) dry etched diaphragms’ cross-sections that can be fabricated using the MEMS-technology-based process in a silicon substrate.

**Figure 2 micromachines-14-01725-f002:**
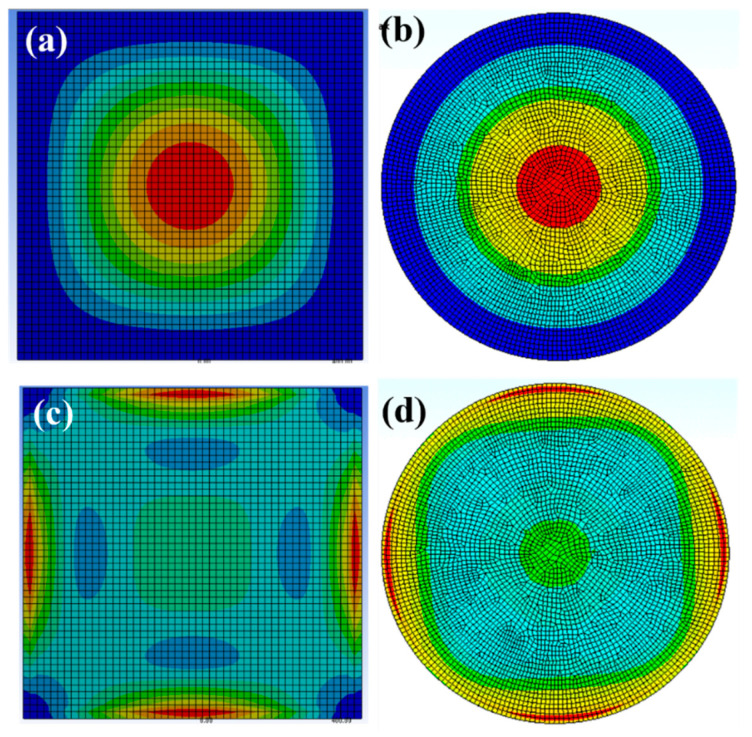
Contour plots for the square and circular diaphragms (**a**,**b**) show deflection (red color is high deflection and blue is low deflection) and (**c**,**d**) induced von misses stress contours (red color is high stress and blue is low stress) of the thin diaphragm under the applied load. The figures demonstrate qualitative visualization of the maximum deflection at the center of the thin diaphragms/plates and induced maximum stresses in the edge of the thin plates.

**Figure 3 micromachines-14-01725-f003:**
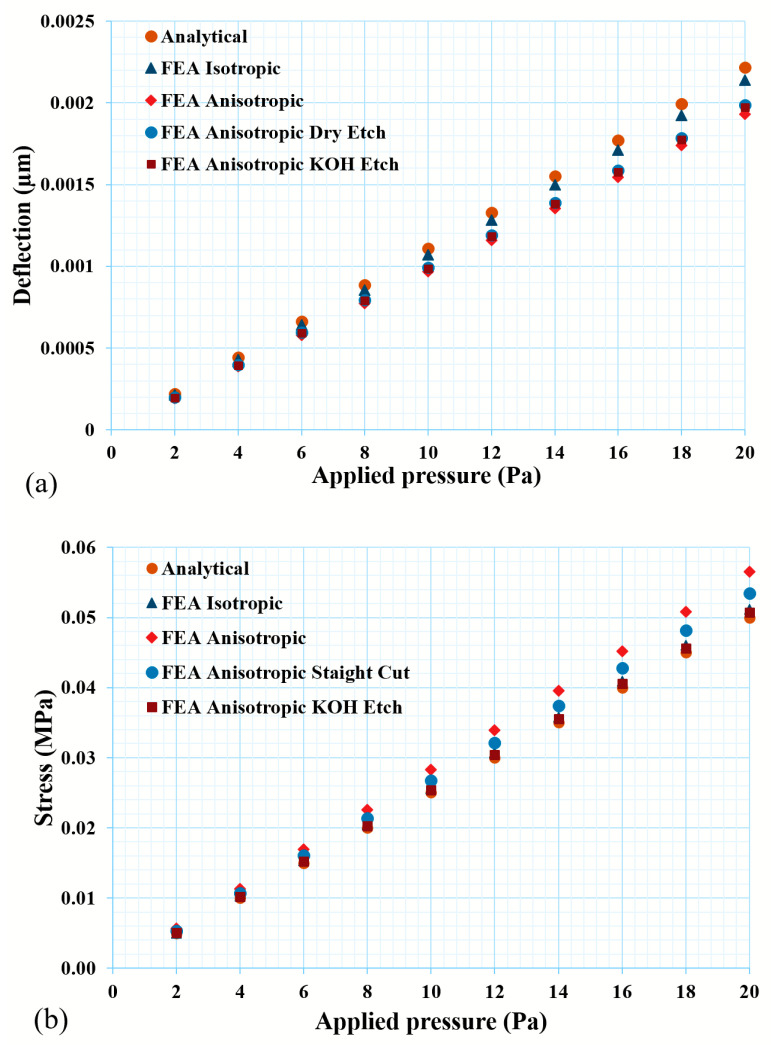
Mechanical characteristics of thin diaphragm: (**a**) maximum deflection and (**b**) induced stress of a thin diaphragm under an applied load of 20 Pa, whose geometrical dimensions are 500 µm × 500 µm × 10 µm.

**Figure 4 micromachines-14-01725-f004:**
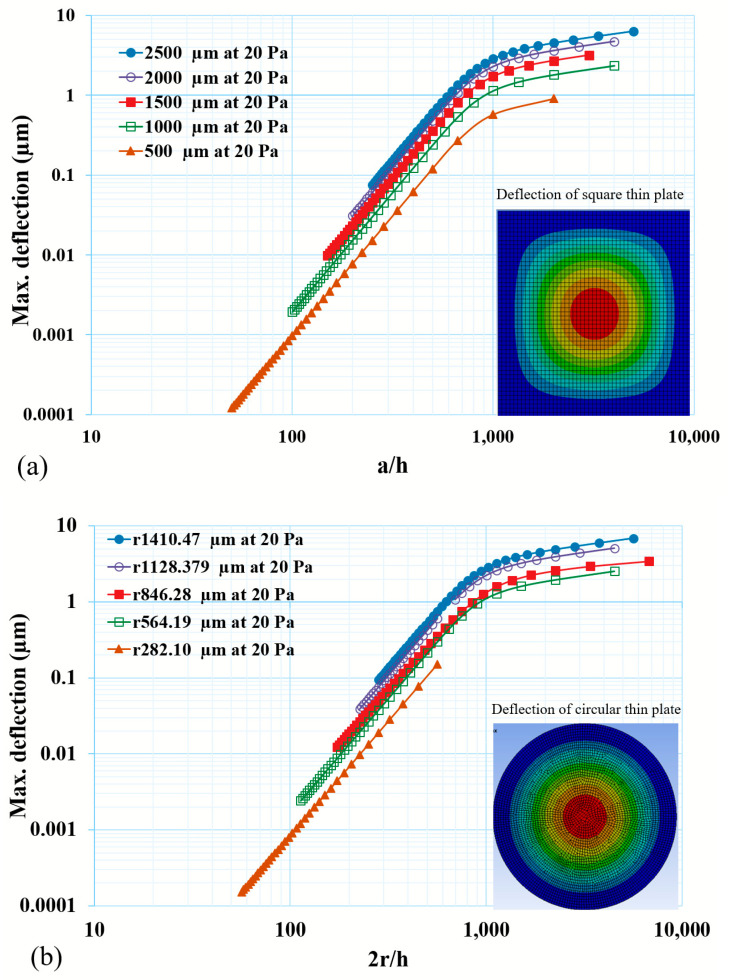
Deflection of a thin diaphragm with varying a/h or 2r/h ratios under 20 Pa applied pressure for (**a**) square and (**b**) circular diaphragms. It suggests that there is a critical limit of the thickness of diaphragm until which the deflection response of thin diaphragm shows high sensitivity and thereafter it decreases because of structural non-linear response.

**Figure 5 micromachines-14-01725-f005:**
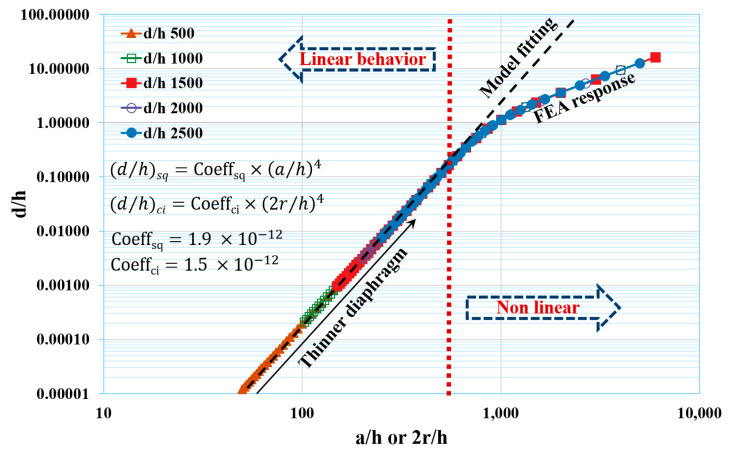
Normalized deflection (*d*/*h*) vs. *a*/*h* or 2*r*/*h* ratio under 20 Pa. The red vertical line is calculated critical values for *a*/*h* or 2*r*/*h* ratio, up to which deflection shows higher deflection sensitivity, thereafter it starts deviating to low sensitive region. Black dashed curve is a fitted model and coefficients for square and circular diaphragms were calculated as 1.9 × 10^−12^ and 1.5 × 10^−12^, respectively. The empirical relations are ${d/h=1.9\times 10^{-12}\times \left(a/h\right)}^{4}$ and ${d/h=1.5\times 10^{-12}\times \left(2r/h\right)}^{4}$ for square and circular diaphragms, respectively.

**Figure 6 micromachines-14-01725-f006:**
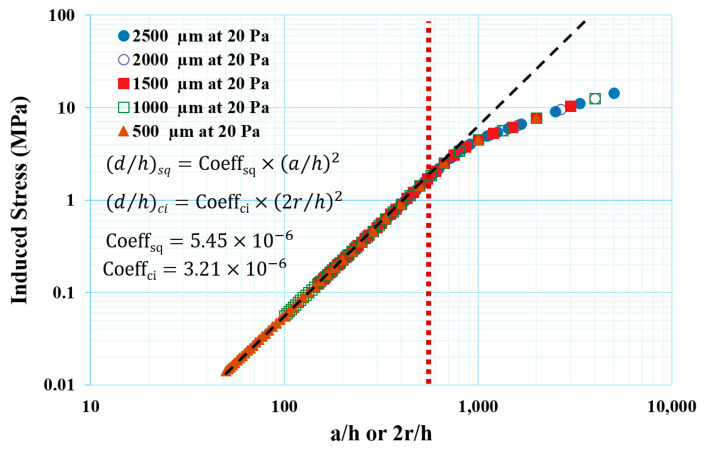
Induced stress vs. a/h ratio of the thin diaphragm under 20 Pa applied pressure. The empirical relations are ${{Stress}_{sq}=5.45\times 10^{-6}\times \left(d/h\right)}^{2}$ and ${{Stress}_{ci}=3.12\times 10^{-6}\times \left(2r/h\right)}^{2}$ for square and circular diaphragms, respectively. Red dotted line is shows a critical ration ad black dashed line is model fitting.

**Figure 7 micromachines-14-01725-f007:**
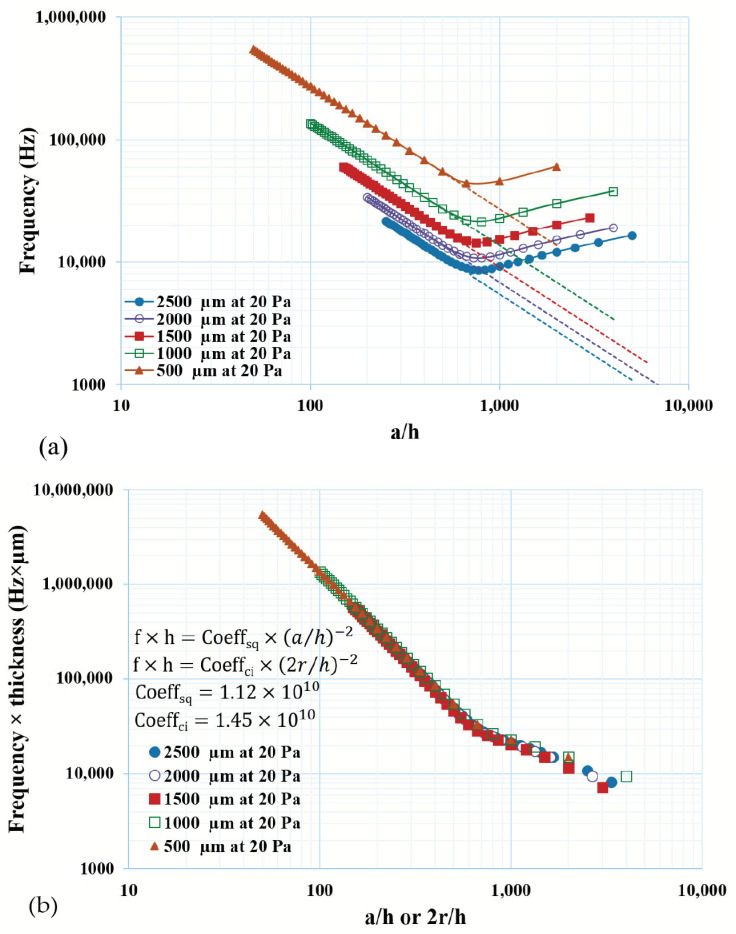
Frequency of thin plate used for microsystems, (**a**) natural frequency of thin square plate as a function of the *a*/*h* ratio and (**b**) scaled frequency through thickness multiplier for thin plate vs. *a*/*h* or 2*r*/*h* ratio, curve fitting is performed, and coefficients were calculated.

**Figure 8 micromachines-14-01725-f008:**
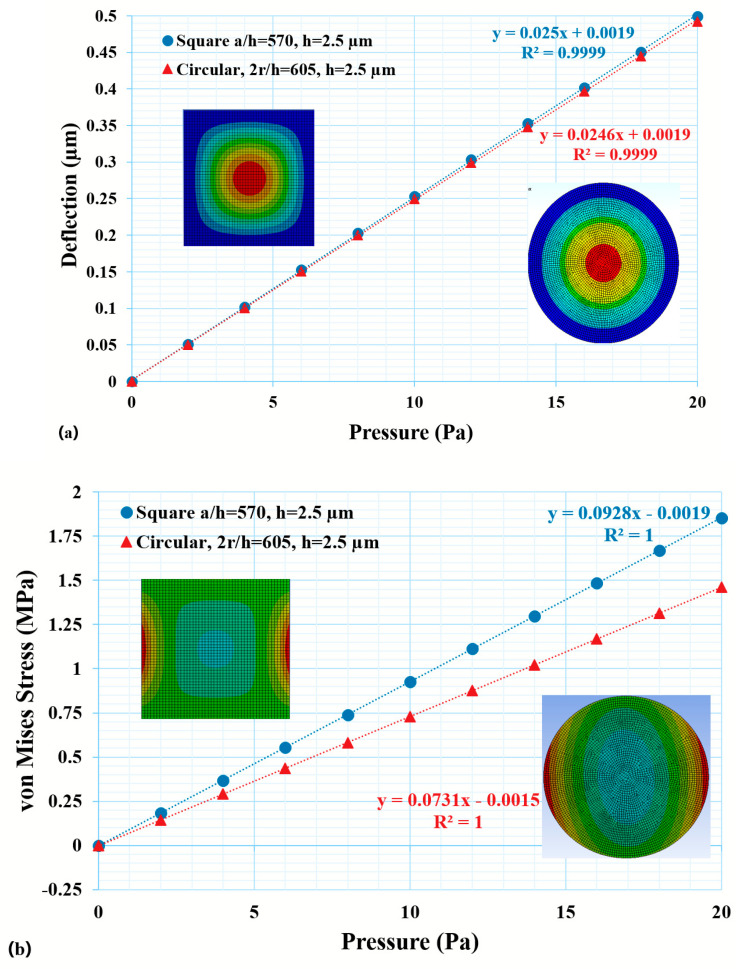
(**a**) Maximum deflection and (**b**) induced direction stress of the thin diaphragms/plates (square and circular). In the figures, the blue and red curves are linear fitting for the square and circular thin diaphragms response under the applied pressure.

**Figure 9 micromachines-14-01725-f009:**
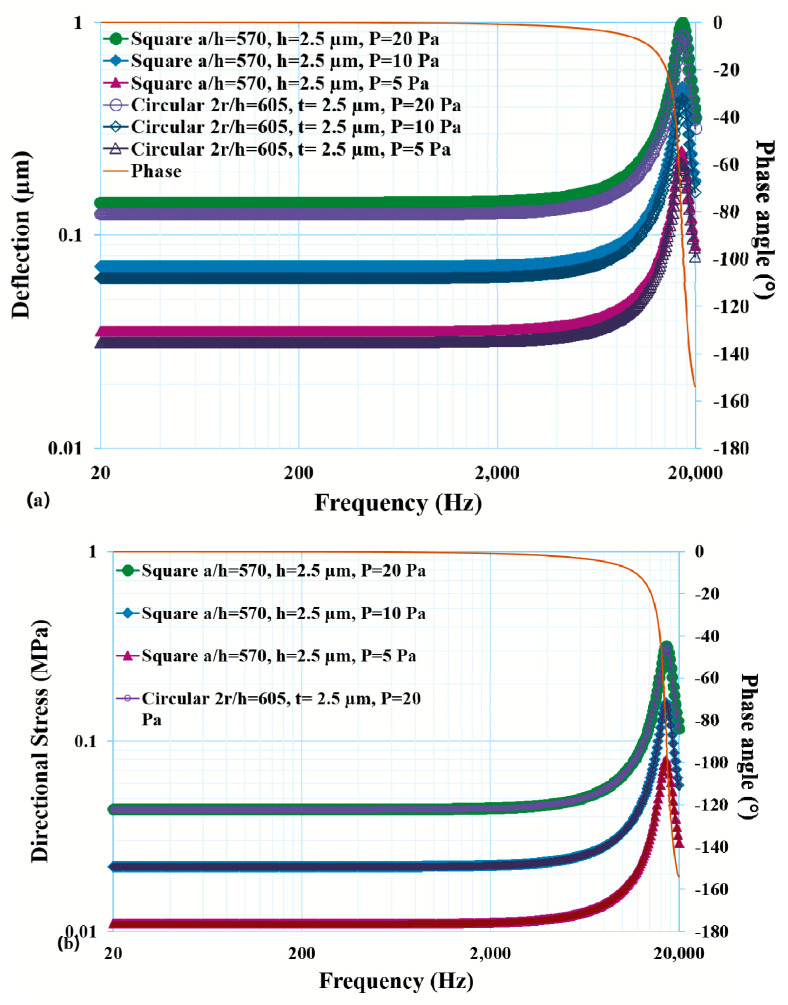
Effect of frequency on (**a**) deflection and (**b**) directional stress with varying pressure.

**Table 1 micromachines-14-01725-t001:** Silicon material properties [43].

Materials	Stiffness Coefficient/Young’s Modulus at 25 °C (GPa)		Poisson’s Ratio
Anisotropic Silicon	c11	165.64	0.27
	c12	63.94	
	c44	79.51	
Isotropic Silicon		169	

**Table 2 micromachines-14-01725-t002:** Deflection and stress sensitivity of a thin diaphragm with geometrical dimensions of 500 µm × 500 µm × 10 µm.

Parameters	Analytical Model	FEA Isotropic	FEA Anisotropic	% Change Isotropic	% Change Anisotropic
Deflection sensitivity (µm/Pa)	1.11 × 10^−4^	1.07 × 10^−4^	9.66 × 10^−5^	−3.48	−12.81
Stress sensitivity (MPa/Pa)	2.50 × 10^−3^	2.56 × 10^−3^	2.83 × 10^−3^	2.27	13.01

**Table 3 micromachines-14-01725-t003:** Established relationship for the deflection, induced stress, and factor of natural frequency coefficients are calculated at 20 MPa pressure.

	Critical Ratio	Deflection Coefficient *d/h = coeff ×* (*a/h*)^4^	Induced Stresses Coefficient *Stress = coeff ×* (*a/h*)^2^	Natural Frequencies *f × h = coeff ×* (*a/h*)^−2^
Square (*coeff_sq_*)	570	1.90 × 10^−12^	5.45 × 10^−6^	1.12 × 10^10^
Circular (*coeff_ci_*)	605	1.50 × 10^−12^	3.21 × 10^−6^	1.45 × 10^10^

**Table 4 micromachines-14-01725-t004:** Deflection and stress sensitivities and frequency of thin plate at optimized *a*/*h* or 2*r*/*h* ratio with assumed thickness of 2.5 µm.

	Critical ratio	Thickness (µm)	Edge or Radius (µm)	Deflection Sensitivity (µm/Pa)	Stress Sensitivity (MPa/Pa)	Frequency (kHz)
Square	570	2.5	1425	0.025	0.105	17.24
Circular	605	2.5	756.25	0.025	0.076	17.27

## Data Availability

The datasets generated during and/or analysed during the current study are available from the corresponding author on reasonable request.

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
