# Peer review of "Design Guidelines for Thin Diaphragm-Based Microsystems through Comprehensive Numerical and Analytical Studies"

_micromachines, 2023, doi:10.3390/mi14091725_

Round 1

Reviewer 1 Report

This work is about the comprehensive guidelines for the design and analysis of thin diaphragms used in various microsystems. This paper used simplified deflection and induced stress to calculate the critical ratio for square and circular diaphragms. This work provided a solution to the MEMS developers to reduce cost and time while conceptualizing TDMS designs. However, some issues should be clarified.

1. The format and marking of the references in this paper are not uniform.

2. The layout of figures and formulas is not uniform.

3. The legend in Figure 2 should be added.

4. In part of Results and Discussion, “This ratio is named as a critical ratio and concluded that if a deflection of a thin diaphragm is beyond the 1/5th of its thickness”. How did the authors reach the above conclusion?

5. The order of the table does not match the description. Please check and correct.

6. The legend in Figure 9 is not clear enough to distinguish the curve corresponding to each parameter.

Author Response

  1. The format and marking of the references in this paper are not uniform.

          It is taken care of and updated wherever is needed.

  1. The layout of figures and formulas is not uniform.

        It is taken care of and updated wherever is needed.

  1. The legend in Figure 2 should be added.

Figure 2 is about visualizing the maximum deflection and induced stresses as a qualitative visualization. Hence, in the caption, it is mentioned, ‘The figures demonstrate qualitative visualization of the maximum deflection at the center of the thin diaphragms/plates and induced maximum stresses in the edge of the thin plates’.

4. In part of Results and Discussion, “This ratio is named as a critical ratio and concluded that if a deflection of a thin diaphragm is beyond the 1/5thof its thickness”. How did the authors reach the above conclusion?

The sentence is rewritten as ‘This ratio is named as a critical ratio and observed that if a deflection of a thin diaphragm is beyond the 1/5th of its thickness”.

  1. The order of the table does not match the description. Please check and correct.

It is taken care of and updated wherever is needed.

  1. The legend in Figure 9 is not clear enough to distinguish the curve corresponding to each parameter.

  It is taken care of and updated wherever is needed.

Reviewer 2 Report

The paper looks sound but it looks more like a review. Should we not define it as a review?

Where is the innovation? This research question needs to be explained more clearly.

The validation is best based by comparing experimental data to modeled one.

No comments

Author Response

  1. The paper looks sound but it looks more like a review. Should we not define it as a review?

It only deals with the TDMS geometry and provides empirical relations to calculate the design parameters of TDMS, hence not defined as a review.

  1. Where is the innovation? This research question needs to be explained more clearly.

The paper explains comprehensive TDMS design rules and provides the empirical relations to calculate the optimised geometrical parameters. (This paper may provide guidelines to beginners/researchers in the diaphragm research area.)

  1. The validation is best based by comparing experimental data to modeled one.

The design guidelines are validated against the analytical values. The comparison with experimental data will be carried out and reported in the future publication.
